# From Overweight to Severe Obesity: Physical Activity and Behavioural Profiles in a Large Clinical Cohort

**DOI:** 10.3390/jfmk10030283

**Published:** 2025-07-24

**Authors:** Francesca Campoli, Elvira Padua, Lucio Caprioli, Saeid Edriss, Giuseppe Annino, Vincenzo Bonaiuto, Mauro Lombardo

**Affiliations:** 1Department of Human Sciences and Promotion of the Quality of Life, San Raffaele Open University, 00166 Rome, Italy; francesca.campoli@uniroma5.it (F.C.);; 2Sports Engineering Laboratory, Department of Industrial Engineering, University of Rome Tor Vergata, 00133 Rome, Italy; lucio.caprioli@uniroma2.it (L.C.); saeid.edriss@alumni.uniroma2.eu (S.E.);; 3Human Performance Laboratory, Centre of Space Bio-Medicine, Department of Medicine Systems, University of Rome Tor Vergata, 00133 Rome, Italy; g_annino@hotmail.com

**Keywords:** obesity, behavioural phenotype, eating behaviour, food preferences, physical activity, sleep quality

## Abstract

**Background:** Behavioural heterogeneity in obesity is increasingly recognised, but how specific dietary patterns, food preferences and physical activity vary between obesity classes remains poorly characterised. **Methods:** We analysed behavioural, dietary, and lifestyle data from 1366 adults attending a tertiary obesity clinic in Italy. Participants were stratified into five obesity classes defined by BMI. Age-adjusted regression models and chi-square tests with Bonferroni correction were used to examine associations between obesity severity and key behavioural outcomes, including food preferences, eating behaviours, physical activity, and self-reported sleep quality. **Results:** The prevalence of uncontrolled eating, skipping meals, and fast eating significantly increased with obesity severity after adjusting for age (all *p* < 0.05). Preference for yoghurt and legumes declined with increasing BMI, whereas preferences for meat and dairy remained stable. Age-adjusted sport participation decreased progressively, with significantly lower odds in Obesity I, II, and IIIA compared to the Overweight group. Sleep quality was highest among overweight participants and declined with obesity severity; night-time awakenings were most frequent in Obesity IIIB. **Conclusions:** Distinct behavioural and lifestyle traits, including lower sport participation, reduced preference for fibre-rich foods, and greater frequency of uncontrolled, fast, and irregular eating, showed overall trends across obesity classes. While these findings suggest the presence of behavioural phenotypes, their interpretation is limited by the cross-sectional design and the use of self-reported, non-validated measures. Future studies should incorporate objective assessments to inform targeted obesity interventions.

## 1. Introduction

The global increase in the prevalence of obesity has underlined the urgent need to understand not only its biological determinants, but also the behavioural and lifestyle patterns that underpin it. As of 2022, 1 in 8 people worldwide were living with obesity, with over 890 million adults affected. Since 1990, global adult obesity has more than doubled, and adolescent obesity has quadrupled [1]. In addition to established metabolic and genetic contributions, it is increasingly recognised that discrete behavioural phenotypes may underlie distinct trajectories of weight gain and resistance to weight loss [2,3]. In particular, the extent to which maladaptive eating patterns, food preferences, and lifestyle behaviours cluster and evolve through increasing degrees of obesity remains poorly characterised [4].

A large body of literature has linked individual behaviours, such as starvation eating, frequent snacking and distracted eating, with altered homeostatic regulation and increased energy intake [5]. Similarly, poor preference for fibre-rich and plant-based foods has been associated with poor diet quality and elevated cardiometabolic risk [6,7]. Although these behaviours are common in people with obesity, the degree of systematic variation according to obesity class is not well defined. The concept of behavioural stratification within obesity, although often invoked, has rarely been studied with the granularity necessary to inform clinical interventions [8]. At the same time, lifestyle factors such as physical activity and sleep quality represent further areas in which behavioural heterogeneity may exist [9].

Physical activity is not only a determining factor in energy balance, but also a modifiable behaviour with important implications for weight management, cardiometabolic risk reduction, musculoskeletal function, and psychological resilience, especially in obese individuals [10]. Despite being a pillar of obesity treatment, structured physical activity is often underestimated, and its decline across different classes of obesity remains poorly quantified.

Obesity is also associated with altered sleep architecture, but it is unclear whether these disturbances follow a linear pattern with increasing adiposity or emerge at specific thresholds [11]. Recent research has focused on the identification of behavioural phenotypes in obesity—defined as recurring patterns of lifestyle traits such as food preferences, eating behaviour, physical activity, and sleep—that may both contribute to and result from excess body weight [12]. A clearer understanding of these trajectories could provide useful information for the development of phenotype-based strategies that more effectively integrate physical activity and other behavioural components as targeted interventions [13].

In this cross-sectional study we examined a spectrum of behavioural and lifestyle characteristics—including eating behaviour, food and taste preferences, self-reported sleep quality, and physical activity—across five BMI-defined obesity classes. Our aim was to delineate the behavioural architecture of obesity across its severity gradient and to assess whether higher degrees of adiposity are associated with consistent changes in preferences, behaviours, and perceived well-being. Such models, if present, may offer a conceptual framework for more individualised and behaviourally anchored approaches in the treatment of obesity.

## 2. Materials and Methods

### 2.1. Subjects

This cross-sectional study was conducted between January 2024 and March 2025 in a tertiary centre for the treatment of obesity in Rome, Italy. All participants were enrolled during scheduled clinical visits as part of an institutional programme for the assessment and management of excess body weight. Recruitment was limited to individuals attending the centre for an initial consultation, ensuring standardised data collection under clinical supervision. Eligible participants were adults aged 18–75 years who provided written informed consent and completed a structured behavioural and lifestyle questionnaire prior to medical assessment. Patients were excluded if they reported a history of cardiovascular or metabolic disease (*n* = 52), if they were pregnant or breastfeeding (*n* = 27), if they had recently participated in a supervised weight loss intervention (*n* = 25), or if they could not provide consistent responses (*n* = 185). No digital or community-based recruitment strategies were used and all data were collected on site. Figure 1 illustrates the flow of participant inclusion and exclusion.

A total of 1366 participants met the inclusion criteria and provided complete data on anthropometric and behavioural variables. Food intake was assessed through structured food diaries, which were only available for participants who returned for a follow-up visit after receiving dietary recommendations at baseline. Therefore, analyses involving food frequency data were limited to this subgroup, while all other outcomes were examined in the entire cohort. Missing data were not imputed. Participants with incomplete responses or no follow-up diary were excluded from specific analyses, as appropriate.

Sample size considerations were based on power estimates performed using G*Power 3.1. Assuming a small to moderate effect size (η^2^ = 0.02), a power of 0.80, an alpha level of 0.05, and five BMI-defined groups, the estimated required sample size was approximately 1600 participants. After applying exclusion criteria, the final sample included 1366 participants. A post hoc power analysis confirmed that this sample size retained a power above 0.78.

The study protocol was approved by the Lazio Area 5 Territorial Ethics Committee (Approval No. 57/SR/23; 7 November 2023) and adhered to the ethical standards outlined in the Declaration of Helsinki.

### 2.2. Questionnaire and Behavioural Assessment

Prior to the clinical assessment, participants completed a structured, self-administered questionnaire designed to capture behavioural, dietary, and lifestyle characteristics relevant to obesity risk and phenotype. The instrument was administered in person at the study site using digital tablets to ensure standardisation and data integrity. Completion time averaged 25–30 min. Informed consent was obtained during the initial clinical meeting and all responses were anonymised at the time of administration to protect the confidentiality of participants. The questionnaire was developed based on previous instruments used in nutritional epidemiology and behavioural research, including those described by Schulz et al. (2021), Hooson et al. (2020), and Carbonneau et al. (2017) [14,15,16]. Although not formally validated, its structure closely followed established instruments and was subjected to internal consistency checks during data cleaning. The section on eating behaviour probed the presence of key patterns involved in the dysregulation of energy intake. Participants were asked about the habit of skipping meals, the speed of eating, eating while distracted or away from the table, and the frequency of episodes characterised by loss of control in the absence of hunger. Additional questions concerned the tendency to eat at night. A separate section dealt with food and taste preferences. The food list was adapted to reflect both the principles of the Mediterranean diet and commonly consumed Western foods, following prior tools developed for Southern European cohorts [17,18]. Finally, participants provided information on structured physical activity, including type, frequency, duration, and preferred time of day. Sports participation was recorded as a dichotomous variable, and weekly activity levels were classified into ordinal bands.

### 2.3. Body Composition

Anthropometric and body composition assessments were conducted on-site using standardised procedures to ensure accuracy and reproducibility. All participants presented themselves fasting (minimum 8 h) and were measured wearing only light undergarments. Body weight was recorded to the nearest 0.1 kg using a calibrated bioelectrical impedance analyser (TANITA BC-420 MA; Tanita Corporation, Tokyo, Japan). Standing height was measured with a wall stadiometer, with the participant positioned according to the Frankfurt horizontal plane. Body mass index (BMI) was calculated as weight in kilograms divided by height in metres squared. Participants were stratified into five BMI-defined categories: Overweight (25.0–29.9 kg/m^2^), Obesity class I (30.0–34.9 kg/m^2^), class II (35.0–39.9 kg/m^2^), class IIIA (40.0–44.9 kg/m^2^), and class IIIB (≥45.0 kg/m^2^). Waist circumference was measured in triplicate at the midpoint between the lowest rib and the iliac crest, with participants standing upright and breathing normally. Body composition parameters, including fat mass (FM), fat-free mass (FFM) and basal metabolic rate (BMR), were obtained using the same bioimpedance device under controlled conditions. To minimise fluctuations related to hydration status or physical exertion, all measurements were taken at least three hours after waking up and food intake and a minimum of 12 h after any strenuous physical activity. Female participants were instructed to avoid measurements during menstruation. Each parameter was measured twice and the average values were used for analysis. Although bioelectrical impedance is widely used in clinical practice due to its ease of use and non-invasive nature, its accuracy in subjects with severe obesity may be limited due to alterations in body water distribution and increased truncal adiposity [19]. For this reason, body composition data were included in the descriptive analyses but were not used as primary outcomes in the inferential models.

### 2.4. Statistical Analysis

Descriptive statistics were calculated for all variables. Continuous data were summarised as means with standard deviations and compared using one-way ANOVA; categorical variables were compared using chi-square tests. Participants were stratified into five BMI categories. For each outcome, pairwise comparisons between BMI classes were performed using post hoc chi-square tests. To control for multiple testing, *p*-values from all pairwise comparisons were adjusted using the Bonferroni method. Age-adjusted logistic regression models were used to evaluate associations between obesity class and binary outcomes, including food preferences, eating behaviours, sleep quality (dichotomized), and sports participation. Ordinal logistic regression was applied to assess sleep quality on a six-point scale. Linear regression was used to analyse sweet–salty taste preference. All models included age as a covariate, and marginal estimates with 95% confidence intervals were reported. No covariates besides age were included in the regression models, as sex, income, and smoking status did not differ significantly between obesity classes and were therefore not considered confounding factors. No subgroup or sensitivity analyses were conducted. Statistical analyses were performed using Python (v3.11).

## 3. Results

The cohort comprised 1366 participants (Table 1), with a predominance of women (56.7%). Age, BMI, waist circumference, fat mass, and BMR increased progressively with obesity severity (all *p* < 0.0001).

Among the 1366 participants, the prevalence of regular consumption (“Yes” response) for low-fat white yoghurt and legumes varied significantly across BMI classes. For low-fat white yoghurt, the proportion of “Yes” responses ranged from 39.4% in Obesity I to 50.0% in Obesity IIIB (*p* = 0.001, chi-square test). For legumes, “Yes” responses ranged from 62.5% in Obesity IIIB to 85.7% in Obesity IIIA (*p* = 0.046, chi-square test). The 95% confidence intervals for each proportion are shown in Figure 2.

No significant differences were observed in sweet, salty, or indifferent taste preference across BMI classes (chi-square *p* = 0.206). This finding remained unchanged after adjustment for age using multinomial logistic regression and after Bonferroni correction for multiple comparisons.

The age-adjusted prevalence of maladaptive eating behaviours across BMI classes is shown in Figure 3 and Appendix A. Uncontrolled eating was highly prevalent and increased with adiposity, from 76.9% in the Overweight group to 100% in class IIIB Obesity (*p* = 0.032). Fast eating showed a non-linear trend, with prevalence ranging from 62.5% to 78.7% (*p* = 0.039). Skipping meals was reported by 31.2% to 43.8% of participants, with significant differences between BMI categories (*p* = 0.034). Other behaviours, including snacking between meals and distracted eating, were frequent across all BMI classes (range: 55.3–81.2% and 62.7–68.8%, respectively) but showed no significant association with adiposity after age adjustment (*p* = 0.105 and *p* = 0.393, respectively). Night eating was less common (14.5–25.7%) and showed a borderline association with BMI class (*p* = 0.062).

Figure 3 reports the age-adjusted percentage of participants who reported skipping meals, uncontrolled eating despite not being hungry, and fast eating, stratified by BMI class. Both uncontrolled eating and fast eating showed significant differences across BMI classes after adjusting for age (*p* = 0.032 and *p* = 0.039, respectively), while skipping meals was also significant (*p* = 0.034).

Sleep quality varied significantly between BMI classes after adjustment for age (Figure 4). The percentage of participants reporting ‘good’ sleep was highest in the Overweight group (52.2%), followed by Obesity II (45.6%) and Obesity I (41.3%). In contrast, Obesity IIIA and IIIB groups reported substantially lower percentages (32.3% and 28.6%, respectively). The frequency of night-time awakenings increased from 37.0% in the Overweight group to 64.3% in Obesity class IIIB, while the prevalence of difficulty falling asleep ranged from 7.1% to 15.2% across BMI classes. The association between BMI class and sleep quality remained significant after adjustment for age (*p* = 0.0038, chi-square test). After Bonferroni correction for multiple comparisons, significant pairwise differences were observed between Obesity I and Overweight (adjusted *p* = 0.045) and between Overweight and Obesity IIIB (adjusted *p* = 0.049).

Sport participation decreased significantly with increasing BMI class (Figure 5). After adjustment for age, the probability of practising sport was highest in the Overweight group (48.8%), followed by Obesity I (37.8%), Obesity II (34.6%), and Obesity IIIA (15.9%). The global association between BMI class and sport participation was highly significant (chi-square *p* < 0.0001). Pairwise post hoc tests with Bonferroni correction confirmed that the probability of practising sport was significantly higher in the Overweight group than in Obesity I (adjusted *p* = 0.0019), Obesity II (adjusted *p* = 0.0025), and Obesity IIIA (adjusted *p* = 0.0009).

## 4. Discussion

This study investigated how behavioural and lifestyle characteristics vary across different classes of obesity, aiming to identify systematic patterns that may reflect distinct behavioural phenotypes. Our findings highlight progressive changes in food preferences, eating behaviours, sleep quality, and physical activity levels with increasing BMI. These results support the hypothesis that behavioural traits cluster differently along the obesity spectrum, with potential implications for personalised intervention strategies.

### 4.1. Food Taste

yoghurt and legumes were the only food items to exhibit significant differences in preference across BMI categories, with markedly lower preference observed among individuals with more severe obesity. These findings contrast with prior epidemiological evidence, which has consistently demonstrated inverse associations between the consumption of these foods and measures of adiposity. Specifically, Eales et al. reported that yoghurt intake was associated with lower body mass index, body weight, and adiposity in both cross-sectional and cohort studies, although randomised trials showed mixed results regarding causality [20]. Similarly, a systematic review by Sayon-Orea and colleagues concluded that yoghurt consumption was linked to favourable weight trajectories and a reduced risk of metabolic syndrome [21]. Prospective analyses from the SUN cohort further supported an inverse relationship between yoghurt consumption and the risk of developing overweight or obesity, particularly when combined with higher fruit intake [22]. For legumes, Tucker found that higher intake was associated with reduced weight gain and lower body fat indices over a 10-year period, although these associations attenuated after adjustment for fibre intake [23]. Heshmatipour et al. reported similar findings in overweight and obese adolescents, with inverse associations between legume consumption and markers of metabolic dysfunction, though statistical significance diminished after full adjustment for confounders [24]. A likely explanation for this discrepancy is that our data reflect declared preferences rather than measured intake. Unlike population-based cohorts that assess actual food consumption, our assessment was conducted before any nutritional counselling, capturing spontaneous patient-reported preferences. These may reflect underlying behavioural traits—such as reduced interest in or aversion to certain healthy foods—that become more pronounced with increasing obesity severity.

### 4.2. Eating Behaviours

Despite extensive evidence linking obesity to taste alterations, our findings did not reveal significant differences in sweet, salty, or indifferent taste preferences across BMI categories. This is somewhat unexpected given that several studies have suggested that individuals with obesity may exhibit altered taste perception or hedonic response to specific flavours, which could influence food choices and energy intake. A recent systematic review concluded that approximately 40% of studies identified significant taste alterations in individuals with obesity, although methodological heterogeneity and limited sensitivity of taste assessments were noted as critical limitations in the current literature [25]. In contrast, maladaptive eating behaviours—including uncontrolled eating, fast eating, and meal skipping—showed significant variation by obesity class. Uncontrolled eating was reported by all patients with class IIIB obesity and was significantly more prevalent than in overweight individuals. These findings align with prior data showing a strong independent association between lack of food control and obesity risk in primary care settings. Rohrer et al. reported that individuals with trouble controlling their eating had a more than sixfold increased odds of being obese compared to those without such difficulties (OR = 6.67; 95% CI, 3.91–11.4) [26]. Fast eating was also more frequent among individuals with obesity, in line with a meta-analysis showing that individuals who eat quickly have significantly higher BMI and greater odds of obesity (OR = 2.15; 95% CI, 1.84–2.51) [27]. Rapid eating may disrupt satiety signalling, leading to increased caloric intake before the onset of fullness. Meal skipping was reported in 31–44% of participants and varied across BMI classes. Recent cohort evidence from over 26,000 Japanese university students suggests that skipping dinner—but not breakfast or lunch—was significantly associated with ≥10% weight gain and the onset of overweight/obesity, with adjusted incidence rate ratios between 1.42 and 1.74 [28]. Together, these findings support the relevance of maladaptive eating behaviours in the progression of obesity and highlight their potential utility in behavioural phenotyping.

### 4.3. Sleep Quality

Sleep quality showed a significant inverse association with obesity class in our cohort. Overweight participants reported sleeping better than individuals in all obesity categories, with those in Obesity IIIA and IIIB showing the highest prevalence of nocturnal awakenings. These findings are in line with previous studies showing that poor sleep duration and quality are associated with an increased risk of obesity [29,30]. Experimental research has shown that sleep restriction increases energy intake by more than 250 kcal per day, largely through alterations in reward processing rather than appetite hormones [31]. Furthermore, higher BMI has been linked to greater sleep fragmentation and lower sleep efficiency, even after adjusting for behavioural and emotional confounders [32]. Although causality cannot be established due to the cross-sectional design, these results suggest that sleep quality may be a relevant component in the behavioural phenotyping of obesity.

### 4.4. Physical Activity

Participants with higher degrees of obesity were less likely to engage in structured physical activity. The most marked drop was observed between Obesity class I and class IIIA, possibly indicating a threshold beyond which participation declines more sharply, although this interpretation remains speculative. These results are consistent with prior evidence suggesting that perceived physical limitations, social stigma, and motivational barriers may contribute to reduced physical activity in individuals with severe obesity [33]. Across all BMI classes, lower activity levels were also associated with older age, underlining the combined effects of obesity and ageing on sedentarism.

### 4.5. Strengths and Limitations

This study has several strengths, including its large sample size, standardised data collection in a clinical setting, and comprehensive assessment of behavioural and lifestyle dimensions. However, some limitations deserve consideration. Nonetheless, several limitations must be acknowledged. First, all behavioural data—including dietary intake, sleep quality, and physical activity—were self-reported, introducing the possibility of recall and social desirability bias. Future studies should employ validated and objective tools such as accelerometers or wearable devices for physical activity and sleep, and dietary apps or supervised 24 h recalls to enhance data accuracy and methodological rigour. Second, body composition was assessed using bioelectrical impedance analysis, which may be less accurate in individuals with severe obesity; therefore, these data were reported descriptively but excluded from inferential models. Third, the cross-sectional design limits causal inference and prevents the identification of temporal relationships. The observed associations may be bidirectional: certain behaviours may contribute to weight gain, but higher adiposity could also reinforce these behavioural patterns. Fourth, although regression models were adjusted for age and post hoc comparisons were corrected for multiple testing, other potential confounders such as sex and income were not included due to incomplete data and limited statistical power for subgroup analyses. As such, residual confounding cannot be ruled out. Fifth, while the behavioural questionnaire was based on previously used instruments and underwent internal consistency checks, it was not formally validated. This represents a methodological limitation; future studies should include psychometric validation and comparisons with gold-standard tools such as the TFEQ, PSQI, or IPAQ to ensure reliability and generalisability. Finally, participants were recruited from a tertiary-level obesity clinic, which may introduce selection bias and limit the generalisability of our findings to broader populations, including those in community or primary-care settings. Future studies should also investigate behavioural differences across sex and age subgroups to identify potentially important modifiers of obesity-related behaviours.

## 5. Conclusions

This study shows that specific behavioural and lifestyle traits cluster with increasing obesity severity (Table 2). Individuals with higher BMI reported reduced sports participation, poorer sleep quality, and lower consumption of foods such as yoghurt and legumes. Fast eating, uncontrolled eating, and meal skipping significantly increased across BMI classes. Recognising these profiles may help tailor clinical interventions to individual behavioural characteristics, enabling more personalised and effective obesity management. Future research should validate these behavioural phenotypes in independent cohorts and assess their utility in guiding personalised obesity treatment strategies.

## Figures and Tables

**Figure 1 jfmk-10-00283-f001:**
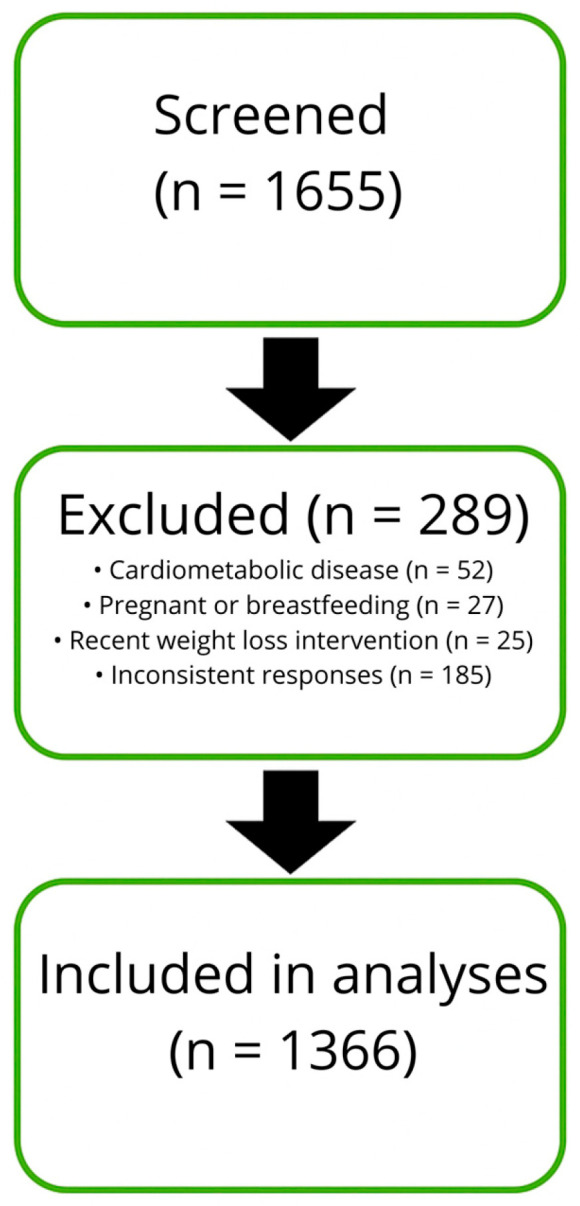
Flow diagram showing the number of participants screened, excluded, and included in the final analyses.

**Figure 2 jfmk-10-00283-f002:**
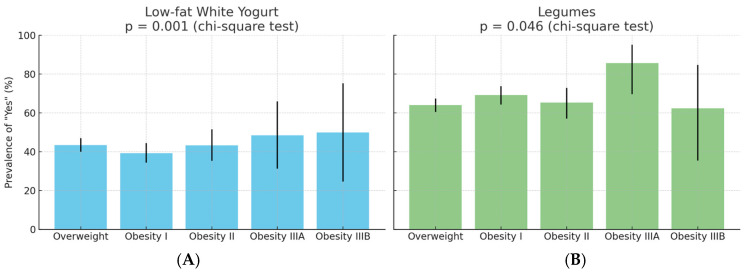
Prevalence of “Yes” responses for (**A**) low-fat white yoghurt and (**B**) legumes by BMI class. Bars indicate the observed percentage of participants in each BMI group reporting regular consumption. Error bars represent 95% binomial confidence intervals. Global comparison by chi-square test: *p* = 0.001 for yoghurt, *p* = 0.046 for legumes.

**Figure 3 jfmk-10-00283-f003:**
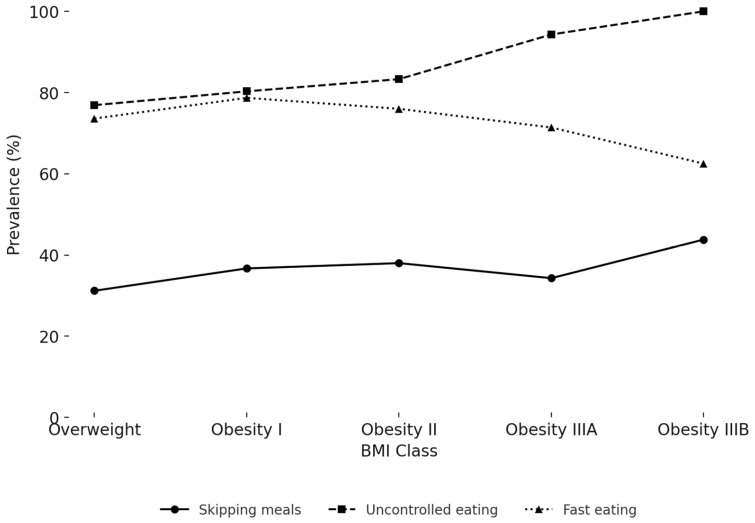
Prevalence of significant eating behaviours by BMI class.

**Figure 4 jfmk-10-00283-f004:**
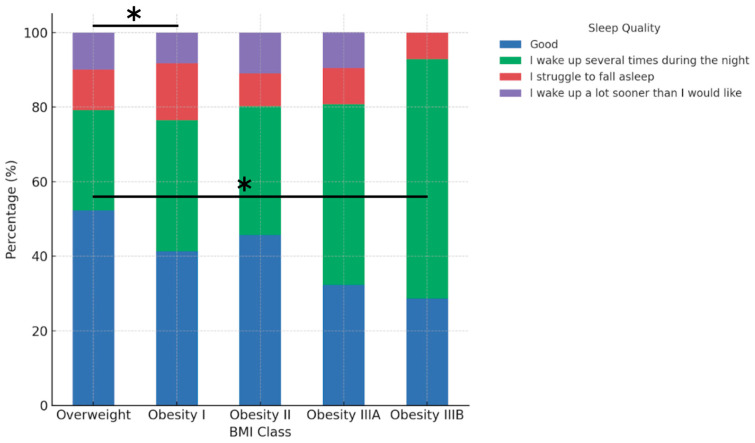
Sleep quality by BMI class. Asterisks indicate statistically significant pairwise differences after Bonferroni correction for multiple comparisons: Overweight vs. Obesity I (*p* = 0.045) and Overweight vs. Obesity IIIB (*p* = 0.049). Global association: chi-square *p* = 0.0038.

**Figure 5 jfmk-10-00283-f005:**
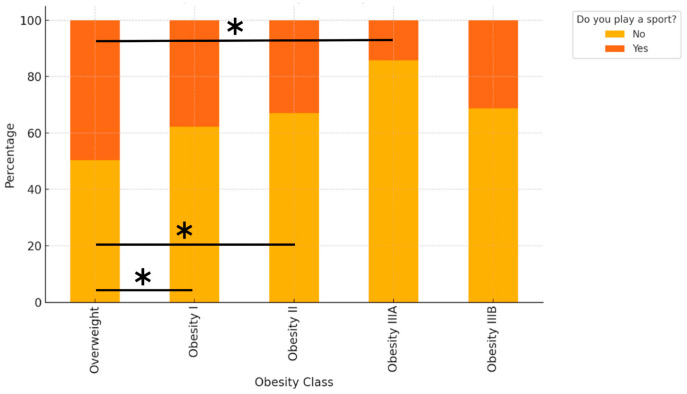
Bars indicate the age-adjusted probability of practising sport (“Yes”) in each BMI class, with 95% confidence intervals. Asterisks indicate statistically significant pairwise differences (Bonferroni-adjusted *p* < 0.01) between the Overweight group and the corresponding Obesity class (I, II, or IIIA). The global association between BMI class and sport participation was highly significant (chi-square *p* < 0.0001). Asterisks indicate significant pairwise differences between Overweight and Obesity I (*p* = 0.0019), Overweight and Obesity II (*p* = 0.0025), and Overweight and Obesity IIIA (*p* = 0.0009), Bonferroni-adjusted.

**Table 1 jfmk-10-00283-t001:** Clinical and socio-demographic characteristics by class of obesity.

	Total (*n* = 1366)	Overweight	Obesity I	Obesity II	Obesity IIIA	Obesity IIIB	*p*-Value
Male (*n*, %)	591 (43.3%)	324 (41.3%)	174 (45.7%)	72 (48.0%)	13 (37.1%)	8 (50.0%)	0.3672
Female (*n*, %)	775 (56.7%)	460 (58.7%)	207 (54.3%)	78 (52.0%)	22 (62.9%)	8 (50.0%)	
Age	42.8 ± 13.2	41.4 ± 13.0	43.2 ± 13.1	46.9 ± 13.3	50.0 ± 12.0	48.6 ± 13.1	<0.0001
Weight (kg)	86.4 ± 16.5	77.6 ± 10.0	92.1 ± 11.3	105.4 ± 13.2	115.5 ± 14.4	143.8 ± 18.2	<0.0001
BMI	30.4 ± 4.7	27.3 ± 1.5	32.2 ± 1.4	37.1 ± 1.4	42.1 ± 1.4	50.9 ± 5.2	<0.0001
FM (kg)	29.4 ± 10.1	23.5 ± 5.2	33.2 ± 6.0	41.4 ± 6.0	51.3 ± 6.4	70.5 ± 14.4	<0.0001
FM (%)	33.8 ± 8.1	30.7 ± 7.1	36.4 ± 7.0	39.7 ± 6.4	44.8 ± 6.1	49.7 ± 7.3	<0.0001
AC (cm)	102.7 ± 11.8	95.9 ± 7.4	107.8 ± 7.1	117.4 ± 8.1	124.8 ± 10.9	143.2 ± 10.3	<0.0001
FFM (kg)	54.0 ± 11.4	51.3 ± 10.2	55.8 ± 11.2	60.7 ± 12.4	61.1 ± 13.6	67.4 ± 12.3	<0.0001
BMR (kcal)	1721.5 ± 345.6	1623.2 ± 295.5	1789.2 ± 329.5	1955.2 ± 381.9	1978.8 ± 397.3	2270.6 ± 396.1	<0.0001

Clinical, anthropometric, and sociodemographic characteristics of participants by BMI class. Continuous values are reported as mean ± SD. Categorical values are given as number and percentage. *p*-values were calculated using one-way ANOVA (continuous variables) and chi-square tests (categorical variables). Abbreviations: BMI, body mass index; BMR, basal metabolic rate; FFM, fat-free mass; FM, fat mass; AC, abdominal circumference.

**Table 2 jfmk-10-00283-t002:** Summary of key findings by obesity class.

	Key Findings
Food Preferences	Lower preference for yoghurt and legumes with increasing obesity class. No relevant differences for meat or other dairy products.
Taste Preference	No association between BMI class and sweet–salty taste preference.
Eating behaviours	Fast eating, uncontrolled eating, and skipping meals increased with BMI class.
Sleep Quality	Overweight participants reported better sleep quality; the proportion reporting “Good” sleep declined and night-time awakenings increased with higher obesity class.
Sports Participation	The Overweight class was more likely to practice sports than all obesity classes.

## Data Availability

This study was registered at ClinicalTrials.gov (ID: NCT06654674). The dataset is publicly available at Mendeley Data: https://data.mendeley.com/datasets/d5sd9zx74d/1 (accessed on 1 July 2025).

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
