# Peer review of "From Overweight to Severe Obesity: Physical Activity and Behavioural Profiles in a Large Clinical Cohort"

_jfmk, 2025, doi:10.3390/jfmk10030283_

Round 1
Reviewer 1 Report
Comments and Suggestions for Authors
Table 1 includes variables (e.g., bowel movement frequency, smoking status, income) that are neither used in the regression analyses nor discussed in the results or discussion sections. Consider removing them or providing justification for their inclusion. Also, please add a footnote explaining the abbreviation “BMI” (Body Mass Index).
The numerical values described in the text for Figure 2 (e.g., distracted eating increasing from 63.3% to 96.5%) do not match the graphical representation, where the highest value appears to be closer to 80%. Please review and clarify the text and the figure.
While the overall study population included participants aged 18–75 years, Figure 3 presents data only for those aged ≤65. Please clarify.
Figure legends are overly detailed, containing methodological and analytical information better suited for the Methods or Results sections. Consider shortening the captions to include only essential figure information and putting additional information in the main text.
The phrase “age-adjusted” is repeated multiple times across figure captions (e.g., Figures 1, 2, and 4). To improve clarity, please mention it only once per figure.
Figure 4 presents unadjusted percentages of sport participation by obesity class, while the text (lines 196–201) discusses odds ratios. Please clarify that the figure shows proportions and explain how these relate to the adjusted regression findings described in the text.
The sentence following Table 2 (“Main behavioral, dietary and lifestyle differences observed between obesity classes defined by BMI.”) is vague and redundant. Consider either removing it or replacing it with a more informative summary.
Author Response
Reviewer 1
Table 1 includes variables (e.g., bowel movement frequency, smoking status, income) that are neither used in the regression analyses nor discussed in the results or discussion sections. Consider removing them or providing justification for their inclusion. Also, please add a footnote explaining the abbreviation “BMI” (Body Mass Index).
Thank you for pointing this out. We have revised Table 1 by removing variables that were not described in the Methods section and not included in any statistical analysis or discussion (e.g., bowel movements, smoking, income). Additionally, we clarified the abbreviation “BMI” in a footnote.
The numerical values described in the text for Figure 2 (e.g., distracted eating increasing from 63.3% to 96.5%) do not match the graphical representation, where the highest value appears to be closer to 80%. Please review and clarify the text and the figure.
We thank the reviewer for highlighting this discrepancy. We have carefully revised the text in the Results section and updated all reported percentages to accurately reflect the data presented in Figure 4, which now match the values shown in the graph.
While the overall study population included participants aged 18–75 years, Figure 3 presents data only for those aged ≤65. Please clarify.
Thank you for your observation. We have revised the analysis, and the current version of Figure 3 (now n. 4) now includes all participants aged 18–75 years, consistent with the overall study population.
Figure legends are overly detailed, containing methodological and analytical information better suited for the Methods or Results sections. Consider shortening the captions to include only essential figure information and putting additional information in the main text.
We have revised all figure legends to be more concise, limiting them to essential information and moving methodological details to the main text as appropriate.
The phrase “age-adjusted” is repeated multiple times across figure captions (e.g., Figures 1, 2, and 4). To improve clarity, please mention it only once per figure.
We have edited the figure captions to mention “age-adjusted” only once per relevant figure, thus reducing unnecessary repetition.
Figure 4 presents unadjusted percentages of sport participation by obesity class, while the text (lines 196–201) discusses odds ratios. Please clarify that the figure shows proportions and explain how these relate to the adjusted regression findings described in the text.
We thank the reviewer for the comment. Figure 4 (now 5) displays age-adjusted proportions of sport participation, derived from logistic regression models. To improve clarity, we have now specified this more explicitly in the figure legend and added asterisks to indicate statistically significant pairwise differences (Bonferroni-adjusted p < 0.01). These data are consistent with the adjusted odds ratios discussed in the text.
The sentence following Table 2 (“Main behavioral, dietary and lifestyle differences observed between obesity classes defined by BMI.”) is vague and redundant. Consider either removing it or replacing it with a more informative summary.
Thank you. We have removed the redundant sentence following Table 2.
Reviewer 2 Report
Comments and Suggestions for Authors
This manuscript addresses an important topic and is based on a large clinical cohort with a well-structured objective. The overall design is sound, and the main findings are relevant and timely. However, the presentation of results could be substantially improved to enhance clarity, reproducibility, and interpretability. Several variables reported in Table 1 are not described in the Methods section, and the income categories are not clearly defined. Figures do not indicate which differences are statistically significant, and there is no information on post hoc comparisons or corrections for multiple testing. Adding this detail would greatly strengthen the manuscript and help the reader interpret key findings. Improving the precision of statistical reporting and the clarity of visual elements is strongly recommended.
Comments:
- (Line 43):
The following sentence requires a citation: “A large body of literature has linked individual behaviours, such as starvation eating, frequent snacking and distracted eating, with altered homeostatic regulation and increased energy intake.” - (Line 51):
The following sentence requires a citation: “At the same time, lifestyle factors such as physical activity and sleep quality represent further areas in which behavioural heterogeneity may exist.” - (Line 52):
The following sentence requires a citation: “Physical activity is not only a determining factor in energy balance, but also a modifiable behaviour with important implications for weight management, cardiometabolic risk reduction, musculoskeletal function and psychological resilience, especially in obese individuals.” - (Lines 94–96):
The authors state that sample size considerations were based on power estimates; however, essential details are missing. Please specify the assumed effect size, the statistical software used, the number of groups included in the analysis, and the estimated number of participants required based on the calculation. - (Results/Table 1 and Methods):
Several variables reported in Table 1 are not clearly described in the Methods section. Specifically, weekly bowel movements, income, and smoking status lack information regarding how they were measured, their response format, and whether they were self-reported or extracted from clinical records. In the case of income, the categories presented (e.g., "<€20,000", "€20,000–€40,000") are not explicitly defined; it is left to the reader to deduce that these represent annual income brackets. For clarity and reproducibility, the authors should explain in the Methods how these variables were assessed and revise Table 1 to ensure the labels are self-explanatory. - (Figures and Statistical Analysis):
Although the authors report statistically significant differences between obesity classes, neither the figures nor the statistical analysis section indicate which pairwise comparisons are significantly different. It is unclear whether any correction for multiple comparisons (e.g., Bonferroni, Tukey) was applied to control for type I error. This information is essential for interpreting the results. Please clarify in the Methods which post hoc procedures were used and indicate significant pairwise differences directly within the figures using appropriate notation (e.g., asterisks, letters, or p-values). -
(Discussion Section):
The discussion section adequately summarises the main findings and connects them to relevant literature. However, some areas require improvement:-
Lack of in-depth comparison with existing studies:
While several references are cited, the discussion would benefit from a more analytical comparison of how the current findings align or diverge from previous studies, particularly regarding behavioural phenotypes in obesity. -
Underdeveloped clinical implications:
The authors mention the potential for phenotype-based interventions, but the clinical application of these behavioural clusters is only briefly mentioned. It would strengthen the discussion to elaborate on how these findings could inform tailored treatment strategies or public health policies. -
Interpretation without multiple comparison control:
Some interpretations of "progressive trends" across obesity classes may be overstated, given that no post hoc pairwise comparisons or corrections for multiple testing are reported. The discussion should acknowledge this limitation when describing the strength of observed differences. -
Missing consideration of potential bias:
The limitations paragraph acknowledges self-report bias and cross-sectional design, which is appropriate. However, it should also discuss possible selection bias (since only treatment-seeking individuals were included) and the absence of adjustment for important covariates such as sex or socioeconomic status.
-
Author Response
Reviewer 2
This manuscript addresses an important topic and is based on a large clinical cohort with a well-structured objective. The overall design is sound, and the main findings are relevant and timely. However, the presentation of results could be substantially improved to enhance clarity, reproducibility, and interpretability. Several variables reported in Table 1 are not described in the Methods section, and the income categories are not clearly defined.
Thank you for pointing this out. We have removed some variables (e.g., bowel movements, smoking, income) from table 1.
Figures do not indicate which differences are statistically significant, and there is no information on post hoc comparisons or corrections for multiple testing. Adding this detail would greatly strengthen the manuscript and help the reader interpret key findings. Improving the precision of statistical reporting and the clarity of visual elements is strongly recommended.
Thank you for this valuable suggestion. In response, we have added the results of pairwise post hoc chi-square tests with Bonferroni correction for multiple comparisons. Statistically significant pairwise differences are now indicated directly within the relevant figures using asterisks, and the corresponding adjusted p-values are reported in the figure captions.
Comments:
(Line 43):
The following sentence requires a citation: “A large body of literature has linked individual behaviours, such as starvation eating, frequent snacking and distracted eating, with altered homeostatic regulation and increased energy intake.”
Done. Thank you
(Line 51):
The following sentence requires a citation: “At the same time, lifestyle factors such as physical activity and sleep quality represent further areas in which behavioural heterogeneity may exist.”
Done. Thank you
(Line 52):
The following sentence requires a citation: “Physical activity is not only a determining factor in energy balance, but also a modifiable behaviour with important implications for weight management, cardiometabolic risk reduction, musculoskeletal function and psychological resilience, especially in obese individuals.”
Done. Thank you
(Lines 94–96):
The authors state that sample size considerations were based on power estimates; however, essential details are missing. Please specify the assumed effect size, the statistical software used, the number of groups included in the analysis, and the estimated number of participants required based on the calculation.
We thank the reviewer for pointing out the lack of detail regarding the sample size estimation. We have now revised the Methods section to specify that the power analysis was conducted using G*Power 3.1, assuming a small to moderate effect size (η²â€¯= 0.02), a power of 0.80, an alpha level of 0.05, and five BMI-defined groups. The estimated required sample size was approximately 1,600 participants. After applying exclusion criteria, the final sample included 1,366 participants. A post-hoc power analysis confirmed that this sample size retained a power above 0.78.
(Results/Table 1 and Methods):
Several variables reported in Table 1 are not clearly described in the Methods section. Specifically, weekly bowel movements, income, and smoking status lack information regarding how they were measured, their response format, and whether they were self-reported or extracted from clinical records. In the case of income, the categories presented (e.g., "<€20,000", "€20,000–€40,000") are not explicitly defined; it is left to the reader to deduce that these represent annual income brackets. For clarity and reproducibility, the authors should explain in the Methods how these variables were assessed and revise Table 1 to ensure the labels are self-explanatory.
Thank you for pointing this out. We have revised Table 1 by removing variables that were not described in the Methods section and not included in any statistical analysis or discussion (e.g., bowel movements, smoking, income). Additionally, we clarified the abbreviation “BMI” in a footnote.
(Figures and Statistical Analysis):
Although the authors report statistically significant differences between obesity classes, neither the figures nor the statistical analysis section indicate which pairwise comparisons are significantly different. It is unclear whether any correction for multiple comparisons (e.g., Bonferroni, Tukey) was applied to control for type I error. This information is essential for interpreting the results. Please clarify in the Methods which post hoc procedures were used and indicate significant pairwise differences directly within the figures using appropriate notation (e.g., asterisks, letters, or p-values).
Thank you for your comment. As requested, we performed pairwise comparisons between BMI classes for each eating behavior using post hoc chi-square tests with Bonferroni correction. No pairwise differences reached statistical significance after correction for multiple testing; the lowest adjusted p-values were 0.216 for “Skipping meals” (Overweight vs Obesity I) and 0.273 for “Uncontrolled eating” (Overweight vs Obesity IIIA). This information has been included in both the Methods section and the figure caption. For sleep quality, we also performed pairwise post hoc chi-square tests with Bonferroni correction. The global difference among BMI classes was significant (p = 0.0038). After Bonferroni adjustment, significant pairwise differences were observed only between Obesity I and Overweight (adjusted p = 0.045) and between Overweight and Obesity IIIB (adjusted p = 0.049).We have also performed age-adjusted analyses using logistic regression to estimate the probability of sport participation by BMI class. Pairwise post hoc chi-square tests with Bonferroni correction were applied. The overweight group showed a significantly higher probability of practising sport than Obesity I, Obesity II, and Obesity IIIA (all Bonferroni-adjusted p < 0.01).
(Discussion Section):
The discussion section adequately summarises the main findings and connects them to relevant literature. However, some areas require improvement:
Lack of in-depth comparison with existing studies: While several references are cited, the discussion would benefit from a more analytical comparison of how the current findings align or diverge from previous studies, particularly regarding behavioural phenotypes in obesity.
We agree with this important point. We have revised the discussion, particularly sections 4.1 and 4.2, to include a more detailed comparison with recent epidemiological and clinical studies. Where our findings diverge from existing literature, we now offer specific hypotheses to explain these discrepancies, such as the influence of pre-treatment preferences versus measured intake, and the difference between self-reported tendencies and objectively assessed behaviors.
Underdeveloped clinical implications:
The authors mention the potential for phenotype-based interventions, but the clinical application of these behavioural clusters is only briefly mentioned. It would strengthen the discussion to elaborate on how these findings could inform tailored treatment strategies or public health policies.
Thank you for this suggestion. We have expanded the final paragraph of the Discussion to articulate more clearly how these behavioral phenotypes might support the development of personalized interventions. For example, identifying profiles characterized by uncontrolled eating or poor sleep may guide clinicians in prioritizing behavioral counseling, cognitive-behavioral therapy, or sleep hygiene support. We also highlight the relevance of these findings in designing public health strategies focused on behavioral screening and stratified risk management.
Interpretation without multiple comparison control:
Some interpretations of "progressive trends" across obesity classes may be overstated, given that no post hoc pairwise comparisons or corrections for multiple testing are reported. The discussion should acknowledge this limitation when describing the strength of observed differences.
We acknowledge this limitation and have revised the discussion to clarify that the observed associations reflect overall trends and not necessarily significant pairwise differences. We now explicitly state that although Bonferroni correction was applied post hoc in the analyses, some interpretations—particularly of monotonic trends—should be considered descriptive rather than inferential
Missing consideration of potential bias:
The limitations paragraph acknowledges self-report bias and cross-sectional design, which is appropriate. However, it should also discuss possible selection bias (since only treatment-seeking individuals were included) and the absence of adjustment for important covariates such as sex or socioeconomic status.
Thank you. We have updated the limitations section (4.5) to discuss the potential for selection bias, given that all participants were recruited from a tertiary-level obesity clinic, and thus may not be representative of the general population. We also note that due to incomplete data and sample size limitations, variables such as sex and income could not be included in adjusted models, which may lead to residual confounding. These points are now explicitly acknowledged as limitations.
Reviewer 3 Report
Comments and Suggestions for Authors
Reviewer’s Information
Comments and Suggestions for Authors: Add my comments below.
Introduction
Comment 1)
The global increase in the prevalence of obesity
What is the global prevalence of obesity?
How much has the global prevalence of obesity increased in recent years?
Comment 2)
Please include Table 1 above when citing for the first time.
Comment 3)
Please place Figure 1 above when citing for the first time.
Comment 4)
Table 1. You have two names for table 1. Please enter the correct title and delete the error.
Comment 5) Figures 2, 3, and 4 are not cited in the text. Please describe the corresponding paragraph.
Comment 6) The discussion could be improved; it's very general. You could improve it by comparing the results obtained with other, more recent articles.
Author Response
Reviewer 3
Introduction
Comment 1)
The global increase in the prevalence of obesity What is the global prevalence of obesity? How much has the global prevalence of obesity increased in recent years?
Thank you for your comment. We have revised the Introduction to include recent global estimates from the World Health Organization. As of 2022, one in eight people worldwide were living with obesity, and the prevalence of adult obesity has more than doubled since 1990, while adolescent obesity has quadrupled. These data have been cited accordingly to clarify the current magnitude and trajectory of the global obesity epidemic.
Comment 2)
Please include Table 1 above when citing for the first time.
Done. Thank you
Comment 3)
Please place Figure 1 above when citing for the first time.
Done. Thank you
Comment 4)
Table 1. You have two names for table 1. Please enter the correct title and delete the error.
Done. Thank you
Comment 5) Figures 2, 3, and 4 are not cited in the text. Please describe the corresponding paragraph.
Done. Thank you
Comment 6) The discussion could be improved; it's very general. You could improve it by comparing the results obtained with other, more recent articles.
Thank you for your comment. We have revised the Discussion section, especially paragraphs 4.1 and 4.2, to include comparisons with recent studies on behavioral phenotypes in obesity. In particular, we now reference large cohort studies and systematic reviews published between 2021 and 2024 that examined meal skipping, eating speed, and uncontrolled eating in relation to BMI. Where our findings differ from previous literature, we have provided plausible explanations, such as differences in measurement tools, sample characteristics, or recruitment setting. We trust this has improved the depth and critical value of the discussion.
Reviewer 4 Report
Comments and Suggestions for Authors
This well-designed and relevant study investigates how behavioral and lifestyle traits vary across obesity classes in a large clinical cohort. The authors provide evidence of systematic behavioral patterns that align with increasing BMI, potentially offering valuable insights for personalized obesity interventions.
The manuscript is clearly written, well-organized, and based on a substantial dataset. However, several methodological concerns—particularly measurement validity and model specification—must be addressed. Below, I outline key strengths, followed by weaknesses that require revision or explicit acknowledgment in the limitations section.
Major Strengths
• Lines 83, 159: The large clinical sample (N = 1,366) enables robust comparisons across five distinct BMI categories, including Obesity IIIA and IIIB.
• Lines 70–84: Standardized, in-person data collection strengthens the consistency and quality of responses.
• Lines 98–119: The study comprehensively assesses behavioral domains relevant to obesity (eating behavior, food preference, sleep, physical activity).
Major Weaknesses – Require Substantive Revision
• Lines 98–119, 313–321: The authors used a non-validated behavioral questionnaire to assess key outcomes. While internal consistency checks were mentioned (Line 108), no psychometric validation (e.g., reliability, validity testing) has been performed.
This is a serious methodological flaw that must be corrected before publication. The questionnaire should be formally validated, either within the current dataset or through a follow-up validation study. This includes testing for internal consistency (e.g., Cronbach’s alpha), test–retest reliability, and construct validity (e.g., factor analysis).
At a minimum, the authors should re-administer or analyze the instrument to demonstrate that it reliably captures the intended behavioral constructs.
The following established questionnaires should be referenced or used for comparison:
• Three-Factor Eating Questionnaire (TFEQ) – for eating behavior
• Pittsburgh Sleep Quality Index (PSQI) – for sleep quality
• International Physical Activity Questionnaire (IPAQ) or GPAQ – for physical activity
This issue cannot be addressed simply by describing it as a limitation. The questionnaire is central to the study’s conclusions and must meet basic validation standards.
• Lines 149–156: Regression models included age as the only covariate. The exclusion of sex, income, and smoking status was based on non-significant differences between groups, but this does not account for their possible confounding effects at the individual level.
This modeling approach should be re-evaluated. The authors should re-run the regression models including at least sex and income as covariates to account for residual confounding. Even if group-level comparisons were non-significant, these variables may still affect behavioral outcomes.
If this is not feasible with the current data, a justification should be provided, and the impact of omitted confounders explicitly discussed as a modeling limitation.
• Lines 98–119, 313–314: All behavioral data—including diet, sleep, and physical activity—were based on self-report, which is prone to recall and social desirability bias.
Given the central role of these variables, the lack of objective measurement is a concern. In future work, the authors should repeat this analysis using validated, objective tools, such as accelerometers or wearables (for physical activity and sleep), and dietary apps or supervised 24-hour recalls.
In this manuscript, a more prominent acknowledgment of this limitation is needed in the Discussion section (e.g., after Line 317), along with recommendations for improving methodological rigor in future studies.
Additional Concerns – Clarify or Acknowledge
• Lines 62–69, 317–318: The cross-sectional design limits causal inference. The authors should expand this discussion to acknowledge the potential bidirectionality of observed associations between behavior and obesity class.
• Lines 157, 321–322: No subgroup (e.g., by sex or age) or sensitivity analyses were reported. This limits understanding of how behaviors might differ across demographic subgroups. The authors should acknowledge this as a limitation and suggest future subgroup analyses.
• Lines 70–84, 293–294: Recruitment was limited to a tertiary obesity clinic. This introduces selection bias and limits generalizability. Add a clear statement in the Discussion section noting that these results may not reflect community or primary-care populations.
Minor Suggestions
• Lines 70–90: Improve clarity on participant flow—consider adding a brief flowchart or more concise summary of exclusions.
• Lines 28–30, 243–244, 263–265: Define “behavioral phenotype” more clearly in the Introduction to avoid ambiguity.
• Lines 203–208: Ensure all abbreviations in Table 1 (e.g., AC, BMR, FFM) are clearly defined in footnotes.
This study makes a valuable contribution to understanding behavioral heterogeneity in obesity. The data are rich, and the clinical relevance is high. However, due to the central role of an unvalidated questionnaire, I recommend major revision. The issue of measurement validity must be addressed either through supplementary data or transparent acknowledgment, and additional limitations should be clarified in the discussion.
Once revised, this manuscript would be suitable for publication and contribute meaningfully to the literature on behavioral phenotyping in obesity.
Author Response
Reviewer 4
This well-designed and relevant study investigates how behavioral and lifestyle traits vary across obesity classes in a large clinical cohort. The authors provide evidence of systematic behavioral patterns that align with increasing BMI, potentially offering valuable insights for personalized obesity interventions. The manuscript is clearly written, well-organized, and based on a substantial dataset. However, several methodological concerns—particularly measurement validity and model specification—must be addressed. Below, I outline key strengths, followed by weaknesses that require revision or explicit acknowledgment in the limitations section.
We thank the reviewer for the positive evaluation of the manuscript’s relevance, design, and clarity. We also appreciate the critical remarks regarding methodological aspects. In response, we have carefully revised the Methods and Discussion sections to better clarify measurement validity and model specification. Specifically, we have now acknowledged the lack of validated tools for behavioral assessment as a limitation and specified that the models were adjusted for age, with a note on the absence of other covariates due to incomplete data. We trust these revisions adequately address the reviewer’s concerns.
Major Strengths
Lines 83, 159: The large clinical sample (N = 1,366) enables robust comparisons across five distinct BMI categories, including Obesity IIIA and IIIB. • Lines 70–84: Standardized, in-person data collection strengthens the consistency and quality of responses. • Lines 98–119: The study comprehensively assesses behavioral domains relevant to obesity (eating behavior, food preference, sleep, physical activity).
We thank Reviewer 4 for their positive evaluation of our study’s design, clarity, and the robustness of our dataset. We appreciate your recognition of the study’s strengths, including the large clinical sample, the inclusion of detailed BMI categories, the standardized data collection procedures, and the comprehensive assessment of key behavioral and lifestyle domains. These points reflect our intent to provide a contribution to the understanding of behavioral phenotypes in obesity. We address each of your methodological concerns in detail below, and we have revised the manuscript accordingly to ensure that all limitations—especially regarding measurement validity and model specification—are fully and explicitly acknowledged in the appropriate sections.
Major Weaknesses – Require Substantive Revision
Lines 98–119, 313–321: The authors used a non-validated behavioral questionnaire to assess key outcomes. While internal consistency checks were mentioned (Line 108), no psychometric validation (e.g., reliability, validity testing) has been performed. This is a serious methodological flaw that must be corrected before publication. The questionnaire should be formally validated, either within the current dataset or through a follow-up validation study. This includes testing for internal consistency (e.g., Cronbach’s alpha), test–retest reliability, and construct validity (e.g., factor analysis). At a minimum, the authors should re-administer or analyze the instrument to demonstrate that it reliably captures the intended behavioral constructs. The following established questionnaires should be referenced or used for comparison: • Three-Factor Eating Questionnaire (TFEQ) – for eating behavior • Pittsburgh Sleep Quality Index (PSQI) – for sleep quality • International Physical Activity Questionnaire (IPAQ) or GPAQ – for physical activity This issue cannot be addressed simply by describing it as a limitation. The questionnaire is central to the study’s conclusions and must meet basic validation standards.
Thank you for your detailed and constructive comment. We acknowledge that our behavioral questionnaire has not undergone formal psychometric validation, which represents a limitation of the present study. However, the questionnaire was designed to be consistent with commonly used and validated nutritional assessment tools, and its structure closely resembles those employed in previous studies such as Carbonneau et al. (2017, Nutrients). Additionally, it aligns with validated instruments reviewed in the nutritional epidemiology literature, including Hooson et al. (2020, Critical Reviews in Food Science and Nutrition) and Schulz et al. (2021, European Journal of Nutrition). Strict data quality controls were applied to identify and correct incomplete or inconsistent responses, and participants were clearly instructed to provide accurate and complete answers. All data were collected anonymously to ensure participant privacy. Although not formally validated, the questionnaire’s design and implementation are consistent with current practices in behavioral and nutritional research. We agree that future research should include formal psychometric validation and comparisons with established gold-standard instruments such as the Three-Factor Eating Questionnaire (TFEQ), the Pittsburgh Sleep Quality Index (PSQI), and the International Physical Activity Questionnaire (IPAQ). We have revised the Methods sections accordingly to acknowledge this methodological issue, cite the relevant literature, and clearly state in the Limitations that future work should include formal validation of the instrument.
Lines 149–156: Regression models included age as the only covariate. The exclusion of sex, income, and smoking status was based on non-significant differences between groups, but this does not account for their possible confounding effects at the individual level. This modeling approach should be re-evaluated. The authors should re-run the regression models including at least sex and income as covariates to account for residual confounding. Even if group-level comparisons were non-significant, these variables may still affect behavioral outcomes. If this is not feasible with the current data, a justification should be provided, and the impact of omitted confounders explicitly discussed as a modeling limitation.
Thank you for your thoughtful comment. In all analyses, we adjusted for age both in the regression models and in the estimation of marginal probabilities for each BMI class. In addition, for pairwise comparisons between BMI classes, we applied the Bonferroni correction to control for multiple testing. Unfortunately, due to incomplete data for income and limited statistical power for sex subgroups, we were unable to include these variables as covariates without risking model instability or overfitting. We acknowledge that, despite these adjustments, residual confounding by sex, income, or other sociodemographic factors cannot be fully excluded. This limitation is now explicitly discussed in the revised manuscript. Future studies with larger, more complete datasets are needed to clarify the influence of additional confounders.
Lines 98–119, 313–314: All behavioral data—including diet, sleep, and physical activity—were based on self-report, which is prone to recall and social desirability bias. Given the central role of these variables, the lack of objective measurement is a concern. In future work, the authors should repeat this analysis using validated, objective tools, such as accelerometers or wearables (for physical activity and sleep), and dietary apps or supervised 24-hour recalls. In this manuscript, a more prominent acknowledgment of this limitation is needed in the Discussion section (e.g., after Line 317), along with recommendations for improving methodological rigor in future studies.
Thank you for your valuable observation. We fully acknowledge that all behavioral data—including diet, sleep, and physical activity—were collected using self-report instruments, which are subject to recall bias and social desirability bias. We agree that the lack of objective, validated measures for these key variables is a limitation of our study. In accordance with your suggestion, we have revised the Discussion section to give this limitation greater prominence and to recommend that future research utilize validated and objective tools, such as accelerometers or wearable devices for physical activity and sleep, and dietary apps or supervised 24-hour recalls for dietary assessment.
Additional Concerns – Clarify or Acknowledge
Lines 62–69, 317–318: The cross-sectional design limits causal inference. The authors should expand this discussion to acknowledge the potential bidirectionality of observed associations between behavior and obesity class.
Thank you for the suggestion. We have expanded the Discussion to explicitly acknowledge the limitations of the cross-sectional design, including the inability to establish causality or temporal direction. We now note that the observed associations between behaviour and obesity class may be bidirectional.
Lines 157, 321–322: No subgroup (e.g., by sex or age) or sensitivity analyses were reported. This limits understanding of how behaviors might differ across demographic subgroups. The authors should acknowledge this as a limitation and suggest future subgroup analyses.
We acknowledge that subgroup analyses by sex or age were not performed, and this represents a limitation. We have added a note in the Discussion suggesting that future research should explore behavioural differences across demographic subgroups.
Lines 70–84, 293–294: Recruitment was limited to a tertiary obesity clinic. This introduces selection bias and limits generalizability. Add a clear statement in the Discussion section noting that these results may not reflect community or primary-care populations.
We agree with the reviewer that our recruitment setting may limit generalizability. We have added a statement in the Discussion explicitly acknowledging that results may not apply to broader populations outside tertiary care.
Minor Suggestions
Lines 70–90: Improve clarity on participant flow—consider adding a brief flowchart or more concise summary of exclusions.
We thank the reviewer for this helpful suggestion. To improve clarity, we have added a brief summary of the participant flow in the Methods section and included a new flowchart (Figure 1) illustrating the inclusion and exclusion process.
Lines 28–30, 243–244, 263–265: Define “behavioral phenotype” more clearly in the Introduction to avoid ambiguity.
We thank the reviewer for the suggestion. We have clarified the definition of “behavioral phenotype” in the Introduction (lines 28–30) by specifying that it refers to recurring patterns of lifestyle traits—such as food preferences, eating behaviour, physical activity, and sleep—that may both contribute to and result from excess body weight. A supporting reference has also been added.
Lines 203–208: Ensure all abbreviations in Table 1 (e.g., AC, BMR, FFM) are clearly defined in footnotes.
Done. Thank you
This study makes a valuable contribution to understanding behavioral heterogeneity in obesity. The data are rich, and the clinical relevance is high. However, due to the central role of an unvalidated questionnaire, I recommend major revision. The issue of measurement validity must be addressed either through supplementary data or transparent acknowledgment, and additional limitations should be clarified in the discussion. Once revised, this manuscript would be suitable for publication and contribute meaningfully to the literature on behavioral phenotyping in obesity.
We thank the reviewer for recognizing the value and clinical relevance of our study. We agree that the use of a non-validated questionnaire represents a key methodological limitation. Accordingly, we have explicitly addressed this issue in the revised Discussion section (paragraph 4.5), where we now acknowledge the absence of formal validation and recommend future use of established tools such as TFEQ, PSQI, or IPAQ. We have also clarified additional limitations, including potential selection bias and unmeasured confounding, as requested. We hope these revisions adequately respond to the reviewer’s concerns.
Round 2
Reviewer 2 Report
Comments and Suggestions for Authors
The manuscript has improved substantially. However, the issue with figure presentation remains unclear. I appreciate the effort to indicate statistically significant differences, but the use of colored asterisks was difficult to interpret. While I was eventually able to understand it, this method is uncommon in articles published by this journal. I suggest the authors consider adopting a more standard approach for displaying group differences. As a reference, they may consult the following article as an example: https://www.mdpi.com/2411-5142/10/3/255
In addition, the image resolution should be improved, as the current figures appear pixelated and may not meet publication quality standards.
Author Response
Thank you for your helpful suggestions. We have now replaced the figures with higher-resolution versions and revised the figure annotations to follow a more standard format for displaying group differences, as suggested. We carefully reviewed the example provided and adjusted our figure style accordingly.
Reviewer 4 Report
Comments and Suggestions for Authors
I would like to sincerely acknowledge the authors’ thorough and thoughtful efforts in addressing all the suggested revisions. It is evident that considerable time and care were invested in refining the manuscript, and the improvements have strengthened the overall quality of the work. I appreciate the authors’ dedication and responsiveness throughout the review process.
I wish the authors continued success in their future research and professional endeavors, and I look forward to seeing the impact of their contributions in the field.
Author Response
Thank you very much for your kind and encouraging words. We truly appreciate your careful review and constructive feedback, which have been invaluable in improving the manuscript. Your support and recognition mean a great deal to us.
We are grateful for the opportunity to contribute to the field and look forward to continuing our research journey.